# The Magnetoelastic Contribution to the Steel Internal Damping

**DOI:** 10.3390/ma16083190

**Published:** 2023-04-18

**Authors:** Antonietta Lo Conte

**Affiliations:** Politecnico di Milano, Department of Mechanical Engineering, Via Giuseppe La Masa 1, 20156 Milan, Italy; antonietta.loconte@polimi.it

**Keywords:** internal damping, magnetoelastic effect, thermodynamic formulation, thermoelastic damping

## Abstract

In this paper, the steel internal damping due to both the thermoelastic and the magnetoelastic phenomena has been investigated through a formulation based on thermodynamical potential joints with a hysteretic damping model. With the aim of focusing on the temperature transient in the solid, a first configuration has been considered, which is characterized by a steel rod with an imposed alternating pure shear strain in which only the thermoelastic contribution was studied. The magnetoelastic contribution was then introduced in a further configuration, in which a steel rod in free motion was subjected to torsion on its ends in the presence of a constant magnetic field. A quantitative assessment of the influence of the magnetoelastic dissipation in steel has been computed according to the Sablik-Jiles model by giving a comparison between the thermoelastic and the prevailing magnetoelastic damping coefficients.

## 1. Introduction

The damping capability represents an important criterion for the selection of the materials for machines and components undergoing vibrations. The internal damping of metallic materials for engineering applications can be ascribed to different physical phenomena.

One of the main causes of the damping is plastic strain but this only occurs at stress levels greater than the usual design values. Other causes of internal damping are the thermoelastic effect and the diffusion of interstitial atoms, which becomes important only beyond a critical frequency, i.e., over the frequency range involved in the vibration typically characterizing mechanical components.

The thermoelastic effect is the basis of thermography, a very common technique for analyzing the stresses in components subjected to time-varying loads. The temperature variations that occur in a homogeneous elastic solid are associated with the in-time variation of the stresses, and, in the case of materials of common engineering interest such as steel and aluminum, they are generally of the order of 0.1 °C ÷ 0.01 °C. Since these temperature values are related to the internal damping characteristics of the material, the heating rate measurement, together with the energy input, was used in the literature, as in [1], for the evaluation of the intrinsic material damping capacity. In particular, recent developments [2,3] show a particular interest in the evaluation of thermoelastic damping in microbeam resonators, as it is an important source of intrinsic energy dissipation and crucial to the design of the micro/nano devices.

The potential of the thermographic technique for the evaluation variation over time of the stress field in a component subjected to cyclic load includes both the ability to describe the physical phenomenon of the local dissipation of energy in the solid due to the thermoelastic effect, which allows one to calculate the temporal variation of the temperature distribution through the Fourier equation, and a high degree of sensitivity in the measurement with infrared techniques of the temperature distribution in the area of interest.

The physical phenomenon of the local dissipation of mechanical energy into thermal energy can be defined through the definition of a constitutive law or based on a thermodynamic analysis of the mechanics of solids.

Finally, in ferromagnetic materials, the magnetoelastic damping significantly contributes to the overall material damping, even at low stress levels and low frequencies. The magnetoelastic aspect of damping has been previously analyzed in the literature, as in [4], in which the damping capacity of highly magnetostrictive Fe-Ga solid-solution alloys was evaluated, or in [5], where the dynamic stability of a soft ferromagnetic beam-plate in a transverse magnetic field was investigated, taking into account the magnetoelastic interaction and magnetic damping. Moreover, in [6], measurements of the magneto-mechanical damping were evaluated in crystalline pure iron, nickel, and cobalt bars, whereas in [7] they were analyzed for a cylindrical permalloy layer subjected to magnetic fields while performing free torsion oscillations. In recent developments, the magneto-mechanical hysteresis damping behavior of Fe-Ga-La alloys has been studied in detail [8] and the damping ratio based on magneto-thermoelasticity considerations was investigated [9] for microbeam resonators.

The magnetoelastic energy dissipation during a stress cycle can be interpreted in terms of two distinct magnetic phenomena: magnetic hysteresis and the formation of local parasite currents [10]. A thorough physical interpretation of these phenomena can be found in the literature, as in [11,12,13], but the treatment of magnetoelastic dissipation from the quantitative point of view is still under investigation.

The treatment from the physical point of view is based on Weiss’ model, considering that a ferromagnetic material is divided into magnetic subdomains, and that, for the magnetic moment, deriving from natural magnetization is close to saturation and is oriented along one of the most easily magnetizable directions of the crystalline structure.

In the absence of an external magnetic field, the magnetic moments of the single domain are casually oriented so that the total magnetic moment is zero or very low. After the application of a stress field, the following occurs:A reversible rotation of the domain along the direction of the applied stress,A reversible displacement of the contour of the domain,A not reversible displacement of the contour of the domain.

The variation in the topology of magnetic domains due to the application of the stress field leads to a condition in which the vector resultant of the overall magnetic moment is still zero. However, a rotation of the magnetic moment of each domain occurs and the consequent magneto-strictive deformation sums up to the purely mechanical deformation and makes the material appear more deformable (∆E effect). The stress-strain curve for low values of stress beyond which the phenomenon achieves saturation has a non-linear trend, and, in the case of time-variable loads, causes dissipation.

From the electromagnetic point of view, the reversible rotation of the domains and the reversible displacement of their contours give rise to a reversible rotation of the magnetic moment of the single domain. The irreversible displacement of domains’ contours causes dissipative phenomena, which, although due to the interaction among the magnetic moments, are generally described as caused by the friction among the contours of the magnetic domains themselves. In the presence of an externally applied constant magnetic field, the application of time-dependent stress due to the dissipative phenomena related to the magnetic interaction among domains gives rise to magnetic hysteresis. The application of stress results, in a way, to being equal to the application of an equivalent (apparent) magnetic field [11]

In this work, the thermodynamic formulation of the magnetoelastic phenomenon is coupled with a thermodynamic formulation of the thermoelastic one [11,14]. This model allows one to estimate the internal damping, both due to thermoelasticity and magnetoelasticity of steels and puts into evidence how, at low values of stress and frequency, the internal damping is mainly of a magnetoelastic nature.

## 2. Thermoelastic Problem

As reported in [15], it is possible to express the thermodynamic potentials that compose the internal energy of the system (Free energy (A); Gibbs potential (G), enthalpy (H)) as functions of only three significant state variables. For example, the temperature (T), the cubic strain ($e_{v}$), and the conventional octahedral tangential components of the strain ($\bar{e}$) can be used to obtain an expression of the Fourier equation for heat transfer capable of providing a quantitative evaluation of the associated thermoelastic effects and the energetic phenomena:(1)$$\beta \nabla^{2}T-{\rho \;c}_{v}\frac{\partial T}{\partial t}-\left[\psi_{1}\frac{\partial e_{v}}{\partial t}{+\psi }_{2}\frac{\partial \bar{e}}{\partial t}\right]\left({T+T}_{0}\right)=0$$
where the specific heat is:(2)$$c_{v}=c_{v}^{*}(T)+\frac{{T+T}_{0}}{\rho }\left[6\alpha \frac{\mathrm{dK}}{\mathrm{dT}}+3\alpha \frac{d^{2}K}{{\mathrm{dT}}^{2}}T\right]e_{v}-\frac{{T+T}_{o}}{\rho }\left[\frac{1}{2}\frac{d^{2}K}{{\mathrm{dT}}^{2}}e_{v}{}^{2}+\frac{16}{3}\frac{d^{2}G}{{\mathrm{dT}}^{2}}{\bar{e}}^{2}\right]$$
(3)$$\psi_{1}=3\alpha \frac{d\left(\mathrm{KT}\right)}{\mathrm{dT}}-\frac{\mathrm{dK}}{\mathrm{dT}}e_{v}$$
(4)$$\psi_{2}=\frac{32}{3}\frac{\mathrm{dG}}{\mathrm{dT}}\bar{e}$$
(5)$$K=\frac{E}{3(1-2\upsilon )}$$

The strains are intended to be measured with respect to the reference configuration, which is characterized by the conditions of zero stress and the base temperature, $T_{0}$ (i.e., T = 0).

The thermodynamic analysis that leads to the Fourier equation for an isotropic elastic solid in the form of Equation (1) explicitly considers the dependence of the elastic modulus on temperature in the expression of the strain energy volumetric density [14]:(6)$$\mathrm{dL}=\bar{\sigma }{\;\mathrm{de}}_{v}+4\;\bar{\tau }\;d\bar{e}$$
where:$\bar{\sigma }$ is the octahedral normal stress,$\bar{\tau }$ is the octahedral tangential stress,$e_{v}$ is the cubic strain,$\bar{e}$ is the tangential octahedral strain.

The dependence of the elastic modulus of the material on the temperature introduces a phase shift between the individual components of the strain tensor and the related components of the stress tensor, which results in the presence of coupling terms between the tensor field of the strains and the scalar field of temperatures. This means that the description of solids stressed with fatigue in an isothermal environment cannot be assigned to the equations of isothermal elasticity when considered separately from the heat equation, but rather it requires the solution of the thermoelastic problem considered as a whole.

It can be observed that the approach based on thermodynamic potential represents, in energetic terms, the phase shift between the components of the stress and strain tensor that, in the works of Zener [16,17] on internal damping, were described through the introduction of a complex component of the elastic modulus.

Equation (1) can also be compared with the equation proposed by Biot [18] for the solution of the thermoelastic problem:(7)$$\beta \;\nabla^{2}{T-\rho \;c}_{v}\;\frac{\partial T}{\partial t}=3\;\alpha \;K\;\left({T+T}_{0}\right)\frac{\partial e_{v}}{\partial t}$$

Biot’s equation can be obtained from Equation (1) by considering the elastic modulus as independent of temperature and interpreting it as an equation of limited validity that is able to describe the problem only as a first approximation.

The variational formulation of the Biot thermoelasticity relations does not allow, in fact, one to realistically include a class of basic problems of the solid mechanics, such as those characterized by zero values in the hydrostatic component of stress or deformation, or in any case small with respect to the values of the deviatoric component of stress or strain.

If the equation proposed by Biot was unconditionally valid, there should be no thermoelastic effect in torsional stress isotropic elastic solids; therefore, the material stressed under these conditions would have zero internal damping.

## 3. Effect of Stress on Magnetic Hysteresis

The application of stress to a ferromagnetic material placed in a magnetic field causes a variation in the intensity of the magnetization that leads to its anhysteretic value [19]. As experimentally revealed, the factors that influence the variation of magnetization are the following:(1)how much lower or higher the magnetization is than its anhysteretic state,(2)how sensitive to the stress the magnetization is when approaching an anhysteretic state,(3)how the anhysteretic magnetization changes with stress.

In order to describe the effect of stress on the magnetic hysteresis, the physical model of Jils and Atherton [20], subsequently extended by Sablik and co-workers [21,22], is considered for polycrystalline magnetic materials when subjected to uniaxial stress (σ) aligned with the magnetic field. Both models refer to a condition in which the applied stress is constant and the magnetic field is variable with time. The model proposed does not consider eddy currents effect, thermal viscosity, or demagnetization, but due to its relative simplicity and physical backgrounds it is one of the most used ones.

In the anhysteretic state which describes thermodynamic equilibrium, a polycrystalline ferromagnetic element is considered as a canonical ensemble of interacting magnetic moments. The effect of the interaction among the domains is represented as a contribution to the magnetic field, so that the effective magnetic field results as:(8)$$H_{e}=\bar{H}+\bar{\alpha }M_{a}{+H}_{\sigma }$$
where $\bar{H}$ is the applied magnetic field, the contribution $\bar{\alpha }M_{a}$ is due to the magnetic interaction among the domains, and the contribution $H_{\sigma }$ is due to the interaction among the domains for the presence of the stress field.

The thermodynamic (or anhysteretic) expression of the magnetization $M_{a}$ is obtained using statistical mechanics by considering the material as being composed of pseudo-domains with defined contours. This hypothesis is compatible with the fact that the points of the anhysteretic curve correspond to the thermodynamic equilibrium status, and every variation on that curve must be reversible; therefore, it cannot involve variations of the contour of the domain. It is:(9)$$M_{a}{=M}_{s}\bar{L}\left(H_{e}/a\right)$$
in which the material constant (a) is partially dependent on the microstructure. The contribution $H_{\sigma }$ can be obtained by considering that, in the case stress and magnetic field are co-axial, we have from the thermodynamics:(10)$$G=U-{\mathrm{TS}+(3/2)\sigma \;\lambda (M}_{a})\;\;U=(1/2)\bar{\alpha }\mu_{0}M_{a}{}^{2}\;{A=G+\mu }_{0}\bar{H}M_{a}$$
and the magnetoelastic energy is expressed as:(11)$$E_{\sigma }=\frac{3}{2}{\sigma \lambda (M}_{a})$$

The effective magnetic field can be consequently written as:(12)$$H_{e}=\frac{1}{\mu_{0}}{\left(\frac{\partial A}{\partial M_{a}}\right)}_{T}=\bar{H}+\bar{\alpha }M_{a}+\frac{3}{2}\frac{\sigma }{\mu_{0}}{\left(\frac{\partial {\lambda (M}_{a})}{\partial M_{a}}\right)}_{T}$$
from which the following can be obtained:(13)$$H_{\sigma }=\frac{3}{2}\frac{\sigma }{\mu_{0}}{\left(\frac{\partial {\lambda (M}_{a})}{\partial M_{a}}\right)}_{T}$$

The irreversible contribution to magnetization deriving from the shift of the contours of the magnetic domains can be described as due to the presence of microstructural obstacles. The magnetic domains, during their displacement, engage with and disengage from such obstacles, giving rise to a dissipative phenomenon (boundaries friction). The irreversible magnetization can be written as:(14)$$M_{i}{=M}_{a}-k\delta^{\prime }\left(\frac{\partial M_{i}}{\partial B_{e}}\right)$$
where δ′= +1 or −1 when $\bar{H}$, respectively, increases or decreases, and in which $B_{e}{=\mu }_{0}H_{e}$.

In Equation (14), the second term with the opposite sign with respect to $\bar{H}$ acts analogously to a mechanical friction term, giving rise to a dissipation. Equation (14) can be rewritten to obtain a differential equation whose solution delivers the irreversible magnetization:(15)$$\frac{\partial M_{i}}{\partial \bar{H}}=\frac{\left(M_{a}-M_{i}\right)}{\frac{\delta^{\prime }k}{\mu_{0}}-\left(\bar{\alpha }+\frac{3\sigma }{{2\mu }_{0}}\left[\frac{\partial^{2}{\lambda (M}_{a})}{\partial M_{a}{}^{2}}\right]\right)\left(M_{a}-M_{i}\right)}$$

From Equation (15), it results that the derivative of the irreversible magnetization is a function of both the deviation of the magnetization from its anhysteretic value (M_a_ − M_i_) and the factor (δ′k) that introduces the irreversibility, and, consequently, the hysteresis. The given model includes a term that considers the bending of magnetic domains between two microstructural obstacles. In addition, this term is a function of the deviation of magnetization from its anhysteretic value (M_a_ − M_i_) through a constant (c), representing the ratio between the initial magnetic susceptibility (not magnetized status) and the related anhysteretic value.

The total magnetization can then be written as the sum of the irreversible magnetization and the last introduced contribution:(16)$${M=M}_{i}+c\left(M_{a}-M_{i}\right)$$

Based on Equation (15), it can be remarked that the presence of a constant applied stress implies a variation of the slope of the hysteretic loop. Such variation is due to the contribution of the stress to the value of the effective magnetic field and is determined by magnetostriction ${\lambda (M}_{a})$.

## 4. Magnetic Hysteresis of a Cylindrical Solid Subjected to Torsion

The Sablik-Jiles model, originally formulated for the case of coaxial stress and a magnetic field, was subsequently extended to the case of a cylindrical solid undergoing constant torsion in the presence of a magnetic field directed along the axis of the cylinder and the time variable (Figure 1a) [23].

From the physical point of view, the application of a couple in the presence of a magnetic field $\bar{H}$ causes a rotation (θ) of the magnetization vector M (Figure 1b). In the case of a material such as steel which shows a positive derivative of the total magnetostriction with respect to magnetization, the magnetization vector rotates towards the direction along which the traction component acts and it leaves the direction along which the compression component acts.

The magnetoelasticity energy is defined as the sum of the contributions due to the traction and the contribution due to compression, both not coaxial with the magnetic field. This can be written as:(17)$$E_{\sigma }=\frac{3}{2}\sigma \left[\lambda \left(M_{a}\left(\sigma \right)\right)\right]\left(\cos^{2}\theta -{v\;\sin}^{2}\theta \right)+\frac{3}{2}\left(-\sigma \right)\left[\lambda \left(M_{a}\left(-\sigma \right)\right)\right]\left(\sin^{2}\theta -{v\;\cos}^{2}\theta \right)$$
where ${\lambda (M}_{a}\left(\sigma )\right)$ is the magnetostriction due to the traction component (*σ*) and ${\lambda (M}_{a}(-\sigma ))$ is the magnetostriction due to the compression component (−*σ*).

Several formulations, both empirical and physical, have been proposed for the function $\lambda \left(M_{a}\right)$. The model under investigation derives the expression of the magnetostriction as a function of magnetization from the minimization of the internal energy with respect to the deformation; in the case of a biaxial stress condition due to torsion, the magnetization is given by:(18)$$\frac{3}{2}\lambda \left(M_{a}\left(\sigma \right)\right)=\frac{\left|b\right|}{b}\left\{{\left[{\left(\frac{2}{3}\frac{b_{\vartheta }\left(\sigma \right)}{Y}\right)}^{2}+\frac{2}{Y}\left(\Phi_{\mathrm{mag}}\left(M_{s}\right)-\Phi_{\mathrm{mag}}\left(M_{a}\left(\sigma \right)\right)\right)\right]}^{1/2}-{\left[{\left(\frac{2}{3}\frac{b_{\vartheta }\left(\sigma \right)}{Y}\right)}^{2}+\frac{2}{Y}\Phi_{\mathrm{mag}}\left(M_{s}\right)\right]}^{1/2}\right\}$$
where b is the isotropic magnetoelastic coupling constant and:(19)$$\Phi_{\mathrm{mag}}\left(M_{a}\left(\sigma \right)\right)=\frac{1}{2}\mu_{0}\alpha {\left[M_{a}\left(\sigma \right)\right]}^{2}$$
$$b_{\vartheta }\left(\sigma \right)=b\left(1+v\right)\left(1-\frac{1}{2}\sin^{2}\vartheta \right)$$
(20)$$b_{\vartheta }\left(-\sigma \right)=b\left(1+v\right)\left(1-\frac{1}{2}\cos^{2}\vartheta \right)$$

Since the trend of $M_{a}\left(\sigma \right)$ with respect to $\bar{H}$ is hysteretic, the behavior of ${\lambda (M}_{a}\left(\sigma )\right)$ with respect to $\bar{H}$ also exhibits hysteresis, but with a symmetric behavior across the λ-axis. The effective magnetic field, acting along the direction of magnetization, is then:(21)$$H_{e}=\bar{H}\cos\left(\pi /4-\vartheta \right)+\bar{\alpha }M_{a}{+H}_{\sigma }$$
where:(22)$$H_{\sigma }=\frac{3}{2}\frac{\sigma }{\mu_{0}}\left[\frac{\partial \lambda \left(M_{a}\left(\sigma \right)\right)}{\partial M_{a}}\left(\cos^{2}\theta -{v\;\sin}^{2}\theta \right)-\frac{\partial \lambda \left(M_{a}\left(-\sigma \right)\right)}{\partial M_{a}}\left(\sin^{2}\theta -{v\;\cos}^{2}\theta \right)\right]$$

Assuming, according to the definition of ${\lambda (M}_{a}\left(\sigma )\right)$ and ${\lambda (M}_{a}\left(-\sigma )\right)$:$$\frac{\partial \lambda \left(M_{a}\left(\sigma \right)\right)}{\partial M_{a}}\to \frac{\partial \lambda \left(M_{a}\left(\sigma \right)\right)}{\partial M_{a}\left(\sigma \right)}$$
(23)$$\frac{\partial \lambda \left(M_{a}\left(-\sigma \right)\right)}{\partial M_{a}}\to \frac{\partial \lambda \left(M_{a}\left(-\sigma \right)\right)}{\partial M_{a}\left(-\sigma \right)}$$
In the considered case, the component of the magnetization acting along the direction of the application of the magnetic field is given by:(24)$$M_{z}=M\;\cos\left(\frac{\pi }{4}-\vartheta \right)$$
and Equation (15) becomes:(25)$$\frac{\partial M_{i}}{\partial \bar{H}}=\frac{\left(M_{a}-M_{i}\right)\cos\left(\pi /4-\vartheta \right)}{\frac{\delta^{\prime }k}{\mu_{0}}-\left(\bar{\alpha }+\frac{3\sigma }{{2\mu }_{0}}\left[\Lambda \left(\sigma \right)\right]\right)\left(M_{a}-M_{i}\right)}$$
where:(26)$$\Lambda \left(\sigma \right)=\frac{\partial^{2}\lambda \left(M_{a}\left(\sigma \right)\right)}{\partial M_{a}{\left(\sigma \right)}^{2}}\left(\cos^{2}\theta -{v\;\sin}^{2}\vartheta \right)-\frac{\partial^{2}\lambda \left(M_{a}\left(-\sigma \right)\right)}{\partial M_{a}{\left(-\sigma \right)}^{2}}\left(\sin^{2}\theta -{v\;\cos}^{2}\vartheta \right)$$

To obtain the angle of rotation of the magnetization with respect to the direction of the applied magnetic field when the cylindrical solid is subjected to torsion, the potential energy is considered as:(27)$${\Omega =-\mu }_{0}\bar{H}M_{a}\cos(\pi /4-\vartheta {)-E}_{\sigma }$$
The condition dΩ/dϑ = 0 delivers the value of ϑ for the configuration of equilibrium corresponding to the magnetic field and to the applied torsional moment. The following is obtained:(28)$$\vartheta =\frac{1}{2}\sin^{-1}\left\{\frac{-1+\sqrt{1+4A^{\prime }}}{2A^{\prime }}\right\}$$
where:(29)$$A^{\prime }=8{\left\{\frac{3}{2}\frac{\sigma \left(1+v\right)}{\mu_{0}\bar{H}M_{a}}\left(\bar{\lambda_{a}}\left(\sigma ,-\sigma \right)\right)\right\}}^{2}$$
(30)$$\bar{\lambda_{a}}\left(\sigma ,-\sigma \right)=\frac{1}{2}\left[\lambda \left(M_{a}\left(\sigma \right)\right)+\lambda \left(M_{a}\left(-\sigma \right)\right)\right]$$

The initial value $(\vartheta_{0})$ of the angle ϑ is a function of the initial anhysteretic magnetization. Subsequently, for increasing torsion torque, ϑ decreases with respect to its initial value.

## 5. Torsional Pendulum with Hysteretic Damping Model

The calculation of the internal damping is carried out for a configuration in which a circular cylindrical solid is inserted in a torsional pendulum device for the measurement of the internal damping and subjected to a constant magnetic field equal to the earth magnetic field and acting along the axis of the cylindrical solid.

The torsional pendulum is a measurement technique that exploits the free vibration of the system made by the solid under testing that corresponds to the torsional stiffness of the system, and a disk corresponding to the torsional moment of the inertia of the system. The configuration is reported in Figure 2.

The internal damping (η), related to a given nth cycle of oscillation evaluated from the evolution of the angular oscillation of the disk can be expressed as a function of logarithmic decrement, as the ratio between two consecutive peaks of the *n*th and *n*th + 1 cycles:(31)$$\delta_{n}=\ln\frac{A_{n}}{A_{n+1}}$$
where $A_{n}$ represents the amplitude of the *n*th peak and A_n+1_ represents the amplitude of the subsequent peak of the oscillation.

The introduction of a dissipative term in Equation (1) allows us to apply the equation to the calculation of the temperature distribution, and, therefore, of the internal damping, of the cylindrical solid with a circular cross section that represents the test sample inserted in a torsional pendulum.

As shown by experimental evidence [24], the internal damping is mainly due to the relaxation phenomenon and is strongly dependent on temperature, frequency, and mechanical hysteresis, which demonstrates a strong dependence on the amplitude and does not show a significant dependence on the frequency.

In the present work, temperature values for which relaxation is an unimportant phenomenon are taken into account and the dissipative term to be introduced in Equation (1) is expressed on the basis of a hysteretic model [25].

This model describes the internal damping as a phenomenon due to irreversible transformations in the elastic solid subjected to cyclic deformations. Furthermore, it provides that the area between the stress-strain curve in the loading phase and the one in the unloading phase is proportional to the dissipated energy and does not depend on the frequency at which the load cycle is covered but depends on the amplitude. The hypotheses of the frequency independence and amplitude dependence are physically confirmed, respectively, by the fact that the stress-strain laws can be assumed to be independent of time and that the dissipation of energy per cycle must necessarily increase as the amplitude of the deformation increases since hysteresis is a non-linear phenomenon.

Given $\bar{\varphi }\;=\;\varphi_{r}\;+\;i\varphi_{i},$ the angle of rotation of the pendulum, and its ${\bar{K}}_{\varphi },$ torsional stiffness, the torsional torque is written as:(32)$$M_{t}={\bar{K}}_{\varphi }\times \bar{\varphi }{=K}_{\varphi }(1+i\chi )\varphi$$
where the axis φ is assumed as the real axis and the parameter χ is the hysteretic damping factor with the condition χ << 1.

The torsional torque $M_{t}$ is then given by the composition of a component in-phase with the angle of rotation ($K_{\varphi }\cdot \varphi$) and of a component 90° out-of-phase ($\chi \cdot K_{\varphi }\cdot \varphi$), which gives rise to a mechanical energy dissipation into thermal energy for a cycle that can be expressed as the sum of a thermoelastic and a magnetoelastic contribution:(33)$$\Delta W=(\chi \cdot K_{\varphi }\varphi_{0}{}^{2}{\pi )}_{\mathrm{thermoelastic}}\;+\;{\Delta W}_{\mathrm{magnetoelastic}}$$

The dissipated energy ΔW corresponds to the area of the hysteresis cycle which, for the assumed damping model, is elliptical with a major axis equal to $\varphi_{0}$ and a minor axis equal to $\chi \cdot K_{\varphi }\cdot \varphi_{0}$.

## 6. Thermoelastic Solution for a Rod Subjected to Alternating Pure Shear Strain

With the aim of analyzing the trend of the temperature transient, the configuration of a steel rod with an imposed alternating pure shear strain was initially analyzed in which only the thermoelastic contribution was studied.

The associated principal strains of the pure shear strain (γ) are:$$e_{1}=\gamma \frac{r}{2R}\sin(2\pi \mathrm{ft})+\alpha T$$
(34)$$e_{2}=\alpha T$$
$$e_{3}=-\gamma \frac{r}{2R}\sin(2\pi \mathrm{ft})+\alpha T$$
where *f* indicates the frequency of the imposed strain cycle. In such conditions, the following is obtained:(35)$$\bar{e}=\sqrt{\frac{2}{3}}\gamma \frac{r}{R}\sin(2\pi f\;t)$$
(36)$$e_{v}=3\alpha T$$

The presence of imposed strains allows us to focus attention on the temperature variation of the cylindrical solid. Consequently, Equation (1) can be expressed as:(37)$$\beta \nabla^{2}T-\left[{\rho c}_{v}+\left({T+T}_{0}\right)\psi_{1}\left(3T\frac{d\alpha }{\mathrm{dT}}+3\alpha \right)\right]\frac{\partial T}{\partial t}{-\psi }_{2}\left({T+T}_{0}\right)\left(\sqrt{\frac{2}{3}}\gamma \frac{r}{R}2\pi f\;\sin(2\pi f\;t)\right)=0$$
with the following boundary conditions:
fort = 0T = 0forr = 0$\frac{\partial T}{\partial r}=0$forr = R$\frac{\partial T}{\partial r}+h_{t}T=0$
where $h_{t}$ is the heat exchange coefficient.

The solution of the problem can be carried out analytically or numerically. In the following, a numerical solution was applied to the case of a cylindrical solid with a diameter of 3.5 mm, 7 mm, and 14 mm in 2¼Cr-1Mo steel whose mechanical and thermal characteristics as a function of temperature, for a range between 25 °C and 300 °C, are reported in Table 1.

The equation for the computation of the temperature distribution in the cylindrical solid subjected to torsion was solved by means of a finite difference numerical method. The numerical solution considers the temperature of the solid as a function of the radial coordinate r and takes into account the dependence on the temperature of the elastic constants of the material and of the quantities that govern the heat exchange. Once the temperature distribution is known, the local entropy density along the radial coordinate r is then calculated from the following equation:(38)$${P(q)=(T+T}_{0})P(s)=\beta \frac{{\left(\mathrm{gradT}\right)}^{2}}{{T+T}_{0}}$$

The material internal damping is expressed through the damping coefficient η:(39)$$\eta =\frac{\Delta W}{\Xi }=2\delta$$
where:ΔW is the mechanical energy dissipated in heat in a period,Ξ is the amplitude of the function describing the elastic energy stored in the volume material into which the dissipation takes place,δ is the logarithmic decrement (Equation (31)).

At local scale, with t* indicating the period of the oscillation, the following holds:(40)$$W=\int_{0}^{t*}P(q)\cdot \mathrm{dt}$$
From Equation (39), it is possible to calculate the local damping coefficient through a simple time integration.

The results relating to the cycle with an imposed strain are shown below, the amplitude of which is defined by a maximum value of the pure shear strain on the contour of the circular section γ_max_ = 1200 × 10^−6^. Once the strain cycle was fixed, two different values of the frequency of the strain cycle equal to 15 Hz and 150 Hz were examined and the absolute temperature of the solid equal to 300 K was assumed.

Figure 3 shows the temperature transient and the entropy production on the section of the cylindrical solid considering the frequency of 15 Hz. The trend, represented for different numbers of cycles, is deeply influenced by the heat exchange at the external surface, whose temperature remains constant and equal to 300 K. The temperature transients and the volumetric density of the entropy production in the node placed on the axis of the rod and in the node placed at a distance from the axis equal to R/2 are reported, respectively, in Figure 4 and Figure 5. A similar thermal transient is observed but the entropy production on the axis of the rod is negligible.

The values of the local internal damping as a function of the radial coordinate obtained production of the cycle are shown in Table 2.

The temperature transient and the entropy production on the section of the cylindrical solid considering the frequency of 15 Hz are reported in Figure 6. Moving from 15 Hz to 150 Hz, a faster thermal transient on the section with higher temperature values can be observed.

With the aim of analyzing the influence of the strain cycle amplitude on the results, the frequency was fixed at 15 Hz; a second value of the imposed strain cycle amplitude (γ_max_ = 600 μγ) was considered in addition to the value considered previously (γ_max_ = 1200 μγ). The trend of the temperature and of the volumetric density of the entropy production on the cylinder section are shown in Figure 7, in which it is possible to observe the significant decrease in both maximum temperature variation and of the entropy production as the amplitude of the strain cycle decreases with cycles.

A numerical experimentation was also conducted for the calculation of the overall damping of the cylinder considering its diameter as variable to evaluate the influence of the dimensions of the specimen on the results obtained. The overall damping values for a cylinder with diameters of 14 mm and 3.5 mm are shown in Table 3 and compared with the results obtained for the diameter of 7 mm previously analyzed.

## 7. Solution for a Rod Subjected to Torsion in Free Motion

With the aim of analyzing the dissipated energy of a thermoelastic nature and for the evaluation of the damping in the oscillation cycles of a magnetoelastic nature, the condition of free motion was considered with the hypothesis of hysteretic damping.

The equation of the free motion of the torsional pendulum, with the hypothesis of hysteretic damping described by the hysteretic damping factor (χ), is written as:(41)$$M_{\varphi }\ddot{\varphi }{+K}_{\varphi }(1+i\chi )\varphi =0$$

The solution presented in [26] for the calculation of the only component of thermoelastic damping is herein reported and can be expressed as:(42)$${\varphi =\varphi }_{0}\exp\{-\omega t{\left[\frac{\sqrt{\left({1+\chi }^{2}\right)}-1}{2}\right]}^{1/2}\}\cdot \cos\{\omega t{\left[\frac{\sqrt{\left({1+\chi }^{2}\right)}+1}{2}\right]}^{1/2}+\phi \}$$
where $\varphi_{0}$ and ϕ are two constants to be determined based on the initial conditions. The logarithmic decrement is given by:(43)$$\delta =\frac{2\;\pi \;\chi }{1+\sqrt{{(1+\chi }^{2})}}≅2\pi \chi$$
having assumed χ << 1.

Considering Equation (42) and assuming as a unit rate the term ${\left(\frac{\sqrt{\left(1+\chi^{2}\right)}+1}{2}\right)}^{1/2}$, according to the hypothesis χ << 1, the strains’ tangential octahedral component can be expressed as:(44)$$\bar{e}=\sqrt{\frac{2}{3\;}}\frac{r}{2}\varphi_{0}\exp\{-\omega t{\left[\frac{\sqrt{\left({1+\chi }^{2}\right)}-1}{2}\right]}^{1/2}\}\cdot \cos\{\omega t+\phi \}$$
where *r* represents the radial coordinate of the circular cross section of the cylinder.

Cubic strain, due solely to the effect of thermal expansion, is instead written as:(45)$$e_{v}=3\alpha T$$

Fourier’s equation for the cylindrical solid with a circular section that oscillates in free motion with a model of hysteretic damping results in:(46)$$\begin{array}{l} \beta \nabla^{2}T-\left[{\rho \;c}_{v}+\left({T+T}_{0}\right)\psi_{1}\left(3T\frac{d\alpha }{\mathrm{dT}}+3\alpha \right)\right]\frac{\partial T}{\partial t}+\\ -\psi_{2}\left({T+T}_{0}\right)\left(\sqrt{\frac{2}{3}}{r\;\varphi }_{0}\;\omega \;\exp\left\{-h\omega t\right\}\left[\left(-\frac{\mathrm{dh}}{\mathrm{dt}}t-h\right)\cos\{\omega t+\phi \}-\sin(\omega t+\phi )\right]\right)=0 \end{array}$$
where the damping ratio (h) is expressed as:(47)$$h={\left[\frac{\sqrt{\left({1+\chi }^{2}\right)}-1}{2}\right]}^{1/2}$$
while the boundary conditions are the following:
fort = 0T = 0forr = 0$\frac{\partial T}{\partial r}=0$forr = R$\frac{\partial T}{\partial r}{+h}_{t}T=0$
where h_t_ is the thermal exchange coefficient and *R* is the radius of the circular section of the cylinder. Equation (46) is solved numerically at finite differences considering the temperature function of the radial coordinate only and the dependence of the elastic constant on temperature and on the constants that regulate the thermal exchange.

The calculation of the contribution ${\Delta W}_{\mathrm{magnetoelastic}}$ in Equation (33) has been carried out according to the model presented in Section 4 but by considering the intensity of the magnetic field oriented along the axis of the cylinder as variable, in accordance with the amplitude of *θ*, and with the corresponding torque moment.

From the numerical point of view, the procedure is as follows. Assuming the initial magnetic field is known, the relative value of the anhysteretic magnetization (*M_a_*) and the initial value of the angle *θ* is calculated. Subsequently, at each time step of Equation (46):
1.The effective magnetic field (*H_e_*) and the variation *d*$\bar{H}$ due to the increment of the applied torque are calculated from the knowledge of the value of *θ* of the previous value of *M_a_* and of the derivatives $\partial \lambda /\partial M_{a}\left(\sigma \right)$ and $\partial \lambda /\partial M_{a}\left(\sigma \right)$;2.A new value of *M_a_* is calculated;3.The variation *dM_i_* is calculated from *θ* and *d*$\bar{H}$ from the previous values of *M_a_* and *M_i_*;4.A new value of *M_i_* is computed from *dM_i_* and from the previous *M_i_*;5.M and the related component *M_z_* are then calculated.6.Steps (1) to (5) are processed until a complete oscillation cycle of M_t_, corresponding to the completion of a cycle of magnetic hysteresis, from which ${\Delta W}_{\mathrm{magnetoelastic}}$ can be evaluated.

The dissipated energy of a thermoelastic and magnetoelastic nature enables the calculation of the damping of the cycle of oscillation applied to the subsequent cycle of the pendulum.

The results of a circular cylindrical solid with 5 mm diameter made of steel 2¼Cr1Mo are presented. The natural frequency of the system is 15 Hz; the amplitude of the initial shear strain is assumed to be equal to 800 μγ, 1000 μγ, and 1200 μγ; and the temperature is 27 °C.

The mechanical and thermal parameters of the considered material are reported in Table 1, considering their dependence on the temperature. The magnetic parameters are reported in Table 4.

Looking at Figure 8, for the first oscillation cycle starting from 1200 μγ, it can be observed that, depending on the applied torque, the angle θ of the axis of magnetization of the solid varies between 30° and 57°. The former corresponds to the maximum torque, in which the principal stresses assume the sign reported in Figure 1a and the latter corresponds to the first inversion of the torque.

In accordance with the model herein proposed, since in Equation (39) both P(s) and Ξ have a different distribution depending on the geometry of the system and on the distribution of stress (or strain), a local damping coefficient and a global damping coefficient of the entire system can be considered for the system in a specific configuration.

The comparison between the damping thermoelastic coefficient and the magnetoelastic coefficient as a function of the radial coordinate is reported in Figure 9 for the stable cycle corresponding to an initial maximum shear strain of 1200 μγ. The magnetoelastic local damping shows a limited variability with the radial coordinate and it is prevailing with respect to the thermoelastic damping. The overall damping of the system is reported in Table 5 for three different values of the initial maximum strain. The overall damping cannot be described in a general way with the damping independent of the examined configuration.

The obtained results confirm that the internal damping is mainly of a magnetoelastic nature. Both contributions to damping increase with the amplitude.

## 8. Conclusions

In the present work, the internal damping due to both thermoelastic and magnetoelastic effects was calculated for a cylindrical solid subjected to torque and inserted into a constant magnetic field.

To analyze the trend of the temperature variation in the solid, it was more appropriate to firstly consider an alternating torsion configuration with imposed sliding rather than the free motion of the solid itself by evaluating exclusively the thermoelastic effect. The results obtained show how the variation of the imposed cycle frequency affected the temperature field on the section of the solid, as an increase in frequency led to a similar trend of the temperature transient for higher temperature values. In addition, worthy of mention is the decrease in the maximum temperature variation of the specimen and in the production of entropy with the decrease in the amplitude of the deformation cycle.

The proposed solution based on thermodynamic analysis and on a model of hysteretic damping is of more general validity than that which would have been obtained with the Biot equation.

This solution, therefore, shows how the internal damping of the material should not be interpreted as a mechanical characteristic, but rather as the result of thermodynamic phenomena strongly dependent on the boundary conditions of the system.

In fact, the numerical calculations based on the proposed solution allow us to obtain an evaluation of the energy dissipated by the irreversible processes, even in the case in question of a solid stressed with alternating torsion, for which the Biot solution would provide a null estimate of the thermal effect.

Given the great importance of the magnetoelastic contribution to damping in materials such as steel, it was necessary to consider a configuration in which the solid was subjected to torsion in free motion. The magnetoelastic contribution to damping was evaluated by adopting the model of Sablik-Jiles, which proved to be valid for the description of the magnetoelastic internal damping in a cylindrical solid subjected to torsion in the presence of a constant magnetic field directed along the axis of the cylinder. The results of the comparison between the damping thermoelastic coefficient and the magnetoelastic coefficient show that the latter is characterized by a limited variability with a radial coordinate and that, as expected, it is prevailing with respect to the thermoelastic damping. These outcomes confirmed that the internal steel damping is mainly linked to the magnetoelastic effect, making the in-depth treatment of the magneto-mechanical damping from a quantitative point of view extremely relevant.

## Figures and Tables

**Figure 1 materials-16-03190-f001:**
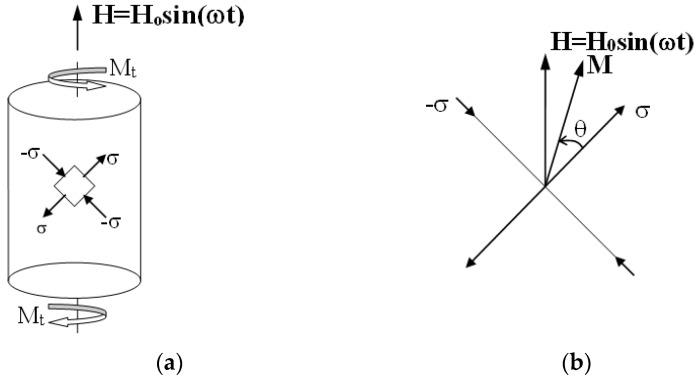
(**a**) Cylindrical solid subjected to torsion, principal stresses, and magnetic field oriented along the axis of the cylinder; (**b**) Relative position of the magnetic vector $\bar{H}$ of the magnetization vector M and of the principal stresses due to the applied torsional moment.

**Figure 2 materials-16-03190-f002:**
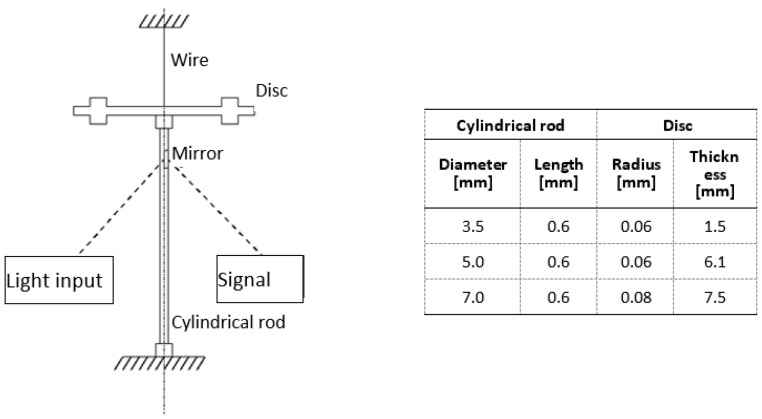
Torsional pendulum for damping measurement (dimensions for *f* = 15 Hz).

**Figure 3 materials-16-03190-f003:**
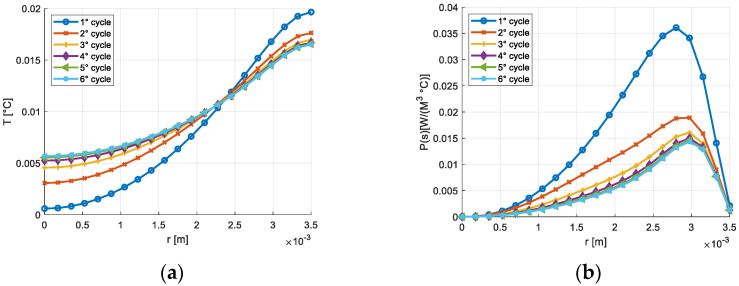
(**a**) Temperature variation on the section of the cylindrical solid, γ_max_ = 1200 μγ, *f* = 15 Hz; (**b**) Entropy production on the section of the cylindrical solid, γ_max_ = 1200 μγ, *f* = 15 Hz.

**Figure 4 materials-16-03190-f004:**
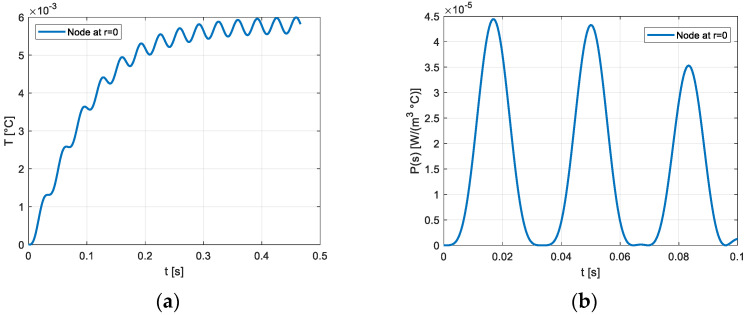
(**a**) Temperature transient in the central node, γ_max_ = 1200 μγ, *f* = 15 Hz; (**b**) Transient of the entropy production on the node adjacent to the central node γ_max_ = 1200 μγ, *f* = 15 Hz.

**Figure 5 materials-16-03190-f005:**
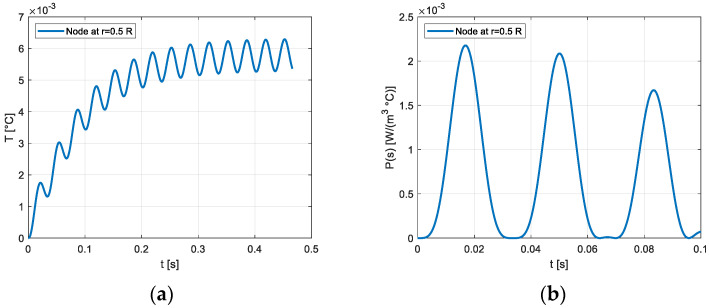
(**a**) Temperature transient on the node located at 0.5 R γ_max_ = 1200 μγ, *f* = 15 Hz; (**b**) Transient of entropy production on the node located at 0.5 R γ_max_ = 1200 μγ, *f* = 15 Hz.

**Figure 6 materials-16-03190-f006:**
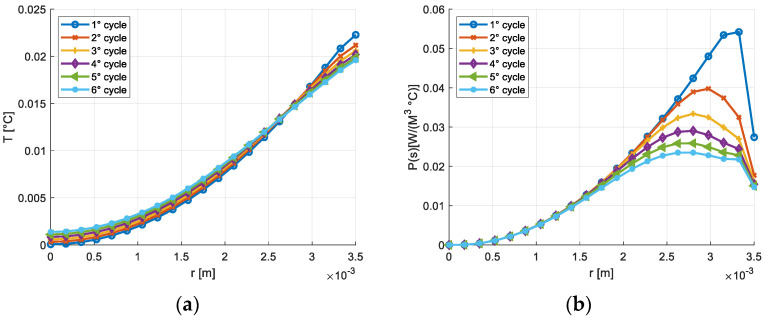
(**a**) Temperature variation on the section of the cylindrical solid, γ_max_ = 1200 μγ, *f* = 150 Hz; (**b**) Entropy production on the section of the cylindrical solid γ_max_ = 1200 μγ, *f* = 150 Hz.

**Figure 7 materials-16-03190-f007:**
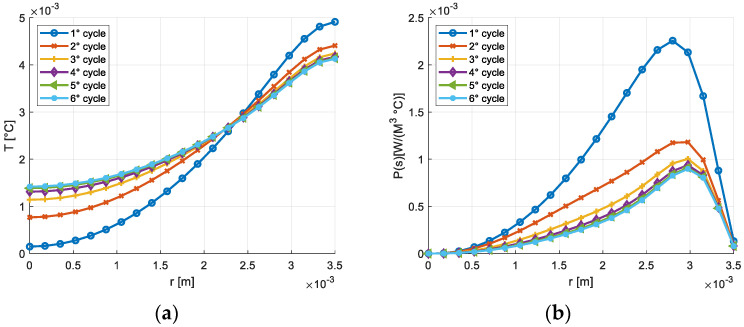
(**a**) Temperature variation on the section of the cylindrical solid, γ_max_ = 600 μγ, *f* = 15 Hz; (**b**) Entropy production on the section of the cylindrical solid γ_max_ = 600 μγ, *f* = 15 Hz.

**Figure 8 materials-16-03190-f008:**
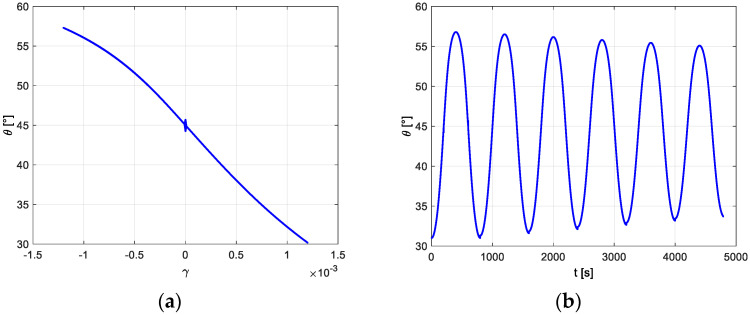
(**a**) Rotation of the magnetization with respect to the maximum strain in the first oscillation cycle. (**b**) Rotation of the magnetization vs. time in the first six cycles.

**Figure 9 materials-16-03190-f009:**
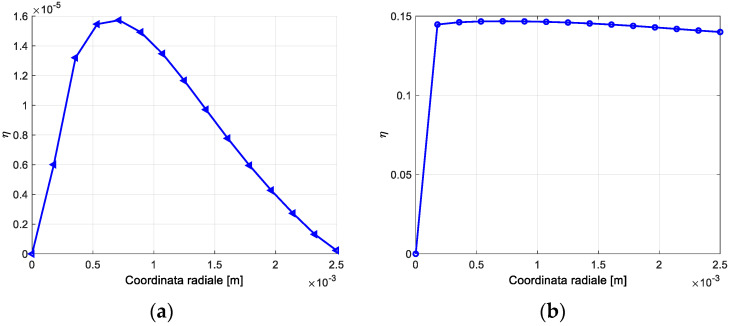
(**a**) Local thermoelastic damping coefficient η as a function of radial coordinate, stable cycle. (**b**) Local magnetoelastic damping coefficient as a function of radial coordinate, stable cycle.

**Table 1 materials-16-03190-t001:** Dependence on temperature of elastic and thermal parameters [14].

E	(215 − 0.085T) × 10^9^	N/mm^2^
$\bar{G}$	(85 − 0.036T) × 10^9^	N/mm^2^
*c_v_*	(425 + 0.416T)	J/(kg °C)
α	(11.3 − 0.014T − 1 × 10^−5^T^2^ ) × 10^−6^	1/°C
β	(35.75 − 0.0223T − 1 × 10^−4^T^2^ )	W/(m °C)
$h_{t}$	10	W/(m^2^ °C)

**Table 2 materials-16-03190-t002:** Local damping coefficients as a function of radial coordinate; values are expressed in 10^−4^.

Node Location	0.1 R	0.2 R	0.3 R	0.4 R	0.5 R	0.6 R	0.7 R	0.8 R	0.9 R
Damping η	0.38	1.06	1.21	1.30	1.32	1.34	1.44	1.66	1.67

**Table 3 materials-16-03190-t003:** Overall damping coefficients as a function of the diameter; values are expressed in 10^−4^.

Diameter (mm)	3.5	7	14
Damping η	3.0	1.3	0.1

**Table 4 materials-16-03190-t004:** Magnetic parameters [23].

$\bar{H}$	36	Asp/m
*M_s_*	1.585 × 10^6^	Asp/m
*λ_s_*	2.07 × 10^−5^	
c	0.15	
a	2350	Asp/m
$\bar{\alpha }$	7.09 × 10^−5^	W/(m °C)
*k/μ* _0_	2400	Asp/m
*μ* _0_	12.57 × 10^−7^	H/m
b	−0.242 × 10^7^	N/m^2^

**Table 5 materials-16-03190-t005:** Internal thermoelastic and magnetoelastic damping coefficients varying the maximum strain amplitude. Values refer to the stabilized cycle.

	800 μγ	1000 μγ	1200 μγ
Thermoelastic damping coefficient **η**	4 × 10^−4^	6 × 10^−4^	9 × 10^−4^
Magnetoelastic damping coefficient **η**	12 × 10^−2^	12.7 × 10^−2^	13.1 × 10^−2^

## Data Availability

The data presented in this study are available on request from the author.

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
