# Peer review of "The Magnetoelastic Contribution to the Steel Internal Damping"

_materials, 2023, doi:10.3390/ma16083190_

Round 1
Reviewer 1 Report
need to make minor corrections as commented in the manuscript.

Author Response
Dear reviewer thank you for the suggestions to improve the quality of the paper.
All the required changes have be done.
Antonietta Lo Conte

Reviewer 2 Report
In this paper, the steel internal damping due to both the thermo-elastic and the magneto-elastic effects has been investigated through a formulation based on thermodynamical potential joints with a hysteretic damping model. A quantitative assessment of the influence of the magneto-elastic dissipation in steel has been computed according to the Sablik and Jiles model by giving a comparison between the thermoelastic and the prevailing magneto-elastic damping coefficient.
This study provides novel and interesting results. In addition, it is very well organized. Meanwhile, the authors should consider the following issues:
-There is a marked revised paper attached below. It provides some required revisions and comments. In addition, there are some typing and grammar errors. The addition of the following recent benchmark study can help to enrich the References part:
-"Analysis of efficiency of passive dampers in multistorey buildings". Journal of Sound and Vibration, 439, 17-28, 2019. https://doi.org/10.1016/j.jsv.2018.09.031
Overall, this paper can be accepted for publication after the completion of the required revisions and comments described above.

Author Response
Dear reviewer thank you for the suggestions to improve the quality of the paper.
1) There is a marked revised paper attached below. It provides some required revisions and comments. In addition, there are some typing and grammar errors.
All the required changes have be done.
2) The addition of the following recent benchmark study can help to enrich the References part: -"Analysis of efficiency of passive dampers in multistorey buildings". Journal of Sound and Vibration, 439, 17-28, 2019. https://doi.org/10.1016/j.jsv.2018.09.031
I thank you for the suggestion to enrich the references part, but as the paper has 27 references almost, and I prefer do not add more.

Reviewer 3 Report
1.What are the advantages and disadvantages of the Sablik and Jiles model, and what are the restrictions when using it?
2.In Figure 1, only torsion is considered. Is it necessary to consider bending moment or shear force in actual reference?
3.Some of the research addressing these issues should be acknowledged, some recommended references, among many others are, https://doi.org/10.1016/j.istruc.2020.12.089. https://doi.org/10.1016/j.jobe.2022.104459. https://doi.org/10.1061/(ASCE)AS.1943-5525.0000942. https://doi.org/10.3311/PPci.15276.
4.The authors considered the relationship between the temperature transient and entropy on the cylindrical solid section at the frequency of 15Hz, and reached the conclusion that with the increase of frequency, there is a faster thermal transient on the section with higher temperature value.Is the frequency sample too small? The correct condition for this conclusion is that the frequency has a specific range. What is the frequency range?Please give clarification.
Author Response
Dear reviewer thank you for the suggestions to improve the quality of the paper.
1.What are the advantages and disadvantages of the Sablik and Jiles model, and what are the restrictions when using it?
The following sentence has been introduced at line 161: "The model proposed does not consider eddy currents effect, thermal viscosity, and de-magnetization, but due to its relative simplicity and physical backgrounds.is one of the most used ones."
2.In Figure 1, only torsion is considered. Is it necessary to consider bending moment or shear force in actual reference?
In this work an thermodynamic energetic formulation of the magnetoelastic phenomenon based on magnetic domains, is coupled with a thermodynamic formulation of the thermoelastic one.
In particular torsional stressed isotropic elastic solids were assumed as examples because there allow to discuss a limitation of the Biot equation for thermoelastic effect.
LIne 146-148 "If the equation proposed by Biot were unconditionally valid, in torsional stressed isotropic elastic solids there should be no thermoelastic effect and therefore the material stressed under these conditions would have zero internal damping."
3.Some of the research addressing these issues should be acknowledged, some recommended references, among many others are, https://doi.org/10.1016/j.istruc.2020.12.089. https://doi.org/10.1016/j.jobe.2022.104459. https://doi.org/10.1061/(ASCE)AS.1943-5525.0000942. https://doi.org/10.3311/PPci.15276.
I thank you for the suggestion to enrich the references part, but as the paper has 27 references almost, and I prefer do not add more.
4.The authors considered the relationship between the temperature transient and entropy on the cylindrical solid section at the frequency of 15Hz, and reached the conclusion that with the increase of frequency, there is a faster thermal transient on the section with higher temperature value.Is the frequency sample too small? The correct condition for this conclusion is that the frequency has a specific range. What is the frequency range?Please give clarification.
The frequency range is 15 Hz-150 Hz and has been reported at line 357.

Round 2
Reviewer 3 Report
The author did not make point to point modifications as suggested by the reviewers, and this manuscript did not meet the requirements of this international journal. It is recommended that the manuscript be rejected.